# Inclusion of chia seeds (*Salvia hispanica* L.) and pumpkin seeds (*Cucurbita moschata*) in dairy sheep diets

**Lizbeth E. Robles Jimenez**[1], **Edgar Aranda Aguirre**[1], **Maria de los Angeles Colin Cruz**[2], **Beatriz Schettino-Bermúdez**[3], **Rey Gutiérrez-Tolentino**[3], **Alfonso J. Chay-Canul**[4], **Ricardo A. Garcia-Herrera**[4], **Navid Ghavipanje**[5], **Octavio A. Castelan Ortega**[1], **Einar Vargas-Bello-Pérez**[6,7]*, **Manuel Gonzalez-Ronquillo**[1]*

1 Facultad de Medicina Veterinaria y Zootecnia, Departamento de Nutricion Animal, Universidad Autonoma del Estado de Mexico, Toluca, State of Mexico, Mexico, 2 Facultad de Quimica, Universidad Autonoma del Estado de Mexico, El Cerrillo Piedras Blancas, Toluca, México, 3 Departamento de Producción Agrícola Animal, Universidad Autónoma Metropolitana Xochimilco, Villa Quietud, Coyoacán, Cd. de México, México, 4 División Académica de Ciencias Agropecuarias, Universidad Juárez Autónoma de Tabasco, Carretera Villahermosa-Teapa, Villahermosa, Tabasco, México, 5 Faculty of Agriculture, Department of Animal Science, University of Birjand, Birjand, Iran, 6 Facultad de Zootecnia y Ecología, Universidad Autónoma de Chihuahua, Chihuahua, Mexico, 7 Department of Animal Sciences, School of Agriculture, Policy and Development, University of Reading, Earley Gate, Reading, United Kingdom

* mrg@uaemex.mx (MGR); e.vargasbelloperez@reading.ac.uk (EV-B-P)

**Data Availability Statement:** All relevant data are within the manuscript and its Supporting

## Abstract

Chia (*Salvia hispanica L.)* seed (CS) and Pumpkin *(Cucurbita moschata)* seed (PS) are used in ruminant diets as energy sources. The current experiment studied the impact of dietary inclusion of CS and PS on nutrient intake and digestibility, milk yield, and milk composition of dairy sheep. Twelve primiparous Texel × Suffolk ewes [70 ± 5 days in milk (DIM); 0.320 ± 0.029 kg milk yield] were distributed in a 4 × 3 Latin square design and fed either a butter-based control diet [CON; 13 g/kg dry matter] or two diets with 61 g/kg DM of either CS or PS. Dietary inclusion of CS and PS did not alter live weight ($p$ >0.1) and DM intake ($p$ >0.1). However, compared to the CON, dietary inclusion of both CS and PS increased the digestibility of neutral detergent fiber ($p$ <0.001) and acid detergent lignin ($p$ < 0.001). Milk production ($p$ = 0.001), fat-corrected milk ($p$ < 0.001), and feed efficiency ($p$ < 0.001) were enhanced with PS, while the highest milk protein yield ($p$ < 0.05) and lactose yield ($p$ < 0.001) were for CS-fed ewes. Compared to the CON diet, the ingestion of either CS and/or PS decreased ($p$ < 0.001) the C16:0 in milk. Moreover, both CS and PS tended to enhance the content of C18:3n6 ($p$ > 0.05) and C18:3n3 ($p$ > 0.05). Overall short-term feeding of CS and/or PS (up to 6.1% DM of diet) not only maintains the production performance and digestibility of nutrients but also positively modifies the milk FA composition.

information files. we have uploaded our data at FIGSHARE: 10.6084/m9.figshare.24424525.

**Funding:** The financial resources for this research was partially provided by Universidad Nacional Autónoma de Mexico (Grant number UAEMex project 4974//2020 CIB) and Universidad Juarez Autónoma de Tabasco. The funders had no role in study design, data collection and analysis, decision to publish, or preparation of the manuscript.

**Competing interests:** The authors have declared that no competing interests exist.

## Introduction

The consumption of foods of animal origin containing saturated fatty acids (SFA) has been linked to an increased propensity for heart and coronary diseases because they can elevate serum levels of total cholesterol, and may raise contents of low-density lipoprotein [1,2]. In this sense, in ruminants, dietary factors can potentially favor the milk fatty acids (FA) profile, making it more suitable for human consumption [reducing SFA and increasing the desirable FA such as isomers of conjugated linoleic acid, especially rumenic acid and vaccenic acid, and α-linolenic acid, as well as improving the n6:n3 ratio] for human consumption [3]. Hence, dietary supplementation with oil seeds as an omega-3 source (i.e., canola, sunflower, linseed, chia seeds, and pumpkin seeds) has been addressed not only used as a nutritional strategy for elevating beneficial FA in milk and milk-based products with no effect on performance or nutrient utilization [2,3] but also as feedstuffs for animal production with less environmental footprints [4,5].

Chia (*Salvia hispanica L.*; a member of the Lamiaceae family) and pumpkin (*Cucurbita moschata*; a member of the Cucurbits family) seeds are from Mexico and have been valued as significant dietary staples since the Mayan and Aztec civilizations. Currently, the annual production of chia seeds (CS) and pumpkin seeds (PS) in Mexico is about 5000 and 6700 tons, respectively [6]. Notably, CS and PS could be used for dairy sheep feeding, due to their high levels of ω-3 and ω-6 FA, as well as soluble and insoluble fibers, proteins, and antioxidants [2,7].

Klir et al. [8] mentioned that the use of PS in dairy goat diets can completely replace soybean meal without decreasing milk production, without abrupt alterations in the FA profile. Similarly, Li et al. [9] fed dairy cows with pumpkin seed cake as an alternative to soybean meal, and that did not affect milk performance, rumen fermentation, and nitrogen excretion. Schettino et al. [10] reported that dietary CS can increase the total content of conjugated linoleic acid (CLA), as well as some CLA isomers such as rumenic acid and linoleic acids in goat milk. Uribe Martinez et al. [11] reported that the inclusion of CS in diets for lambs did not increase carcass weight and did not affect the chemical composition of meat. However, it tended to increase oleic acid and decrease stearic acid in meat (*Longissimus thoracis*). Park et al. [12] determined the effect of including flax seeds plus chia seeds in dairy cows' diets and reported that CLA and polyunsaturated fatty acid (PUFA) contents in milk increased with the supplementation of flaxseed plus chia seeds.

Chia seed contains between 30–40 g/100g total lipids with a PUFA content that can be transferred to milk [13]. It is also a rich source of fiber (30–34 g/100g), proteins (16–24 g/100g), antioxidants, and vitamins (notably B vitamins) as well as minerals (six times higher calcium, eleven times greater phosphorus, and four times larger potassium content compared to milk) [10].

Pumpkin seed has a crude protein content of approximately 35 g/100g and from 30 to 50 g/100g of oil [2].

However, there is scarce data on the inclusion of dietary CS or PS for ewe on production performance, digestibility, nitrogen balance, and milk FA profile. We hypothesized that the inclusion of CS and PS can be applied as a high-quality FA source that could provide a functional role in the milk yield from dairy sheep. To test this hypothesis, an investigation was conducted to evaluate the impacts of CS and/or PS ingestion on nutrient intake and digestibility, nitrogen balance, milk production, and milk FA composition in sheep along with an *in-vitro* gas production (IVGP) test. The data obtained from this research hold significant relevance for sheep farmers who are seeking alternative energy and protein sources that not only meet the growing demand of the market but also reduce consumers' concerns about the healthiness of dairy products.

## Material and methods

### Ethics statement

This research was conducted following the protocols set by the Professional Committee on Standardization of Experimental Animals of the Universidad Autónoma del Estado de México and Universidad Juárez Autónoma de Tabasco.

### Experimental design, animals, and housing

Twelve primiparous Texel × Suffolk ewes [70 ± 5 days in milk (DIM), 0.320 ± 0.029 kg milk yield, (mean ± SD)] were fed three experimental diets and distributed in a 4 × 3 Latin square design, with periods of 20 days. During each period, there were 15 days allocated for diet adaptation, followed by 5 days designated for sample collection. Throughout the study, animals were individually housed in metabolic cages (1.2 m × 0.8 m) and provided with daily feeding (twice at 08:00 and 15:00 hours) for *ad libitum* consumption and continuous water supply. The ewes were hand milked in their standings, once daily (at 16:00 h).

### Experimental diets

Diets were isocaloric [11.5 MJ metabolizable energy (ME) / kg DM] and isonitrogenous [142 g crude protein (CP) / kg DM] to meet the energy and protein requirements of mid-lactation ewes [14]. Table 1 shows the chemical composition of the ingredients. Three dietary treatments were as follows: butter-based (INCA®; ACH Foods, México) control diet (CON) which contained edible tallow, hydrogenated vegetable fat, and tertiary butylhydroquinone (TBHQ)

**Table 1. Chemical composition (g/kg DM) of dietary ingredients in lactating dairy ewes.**

| item | Corn silage | Chia Seed | Pumpkin Seed | SBM | Sorghum grain | Bran wheat | Oats hay | Butter INCA® |
|---|---|---|---|---|---|---|---|---|
| DM † | 286 | 966 | 927 | 893 | 945 | 870 | 897 | 990 |
| OM | 949 | 934 | 967 | 929 | 984 | 930 | 927 | 990 |
| CP | 78.2 | 290 | 290 | 525 | 80.0 | 120 | 120 | 0.0 |
| EE | 51.9 | 270 | 230 | 85.5 | 27.0 | 45.0 | 13.0 | 987 |
| NDF | 439 | 558 | 404 | 63.6 | 60.0 | 456 | 534 | 0.0 |
| ADF | 270 | 285 | 242 | 15.0 | 43.0 | 135 | 313 | 0.0 |
| ADL | 2.5 | 199 | 168 | 3.20 | 3.18 | 2.53 | 92.0 | 0.0 |
| Ca | 2.8 | 6.31 | 0.29 | 3.5 | 4.0 | 1.3 | 3.5 | 0.0 |
| P | 2.3 | 8.6 | 0.71 | 1.9 | 3.2 | 12.9 | 4.5 | 0.0 |
| ME, Mj/kg DM | 11.5 | 15.0 | 13.0 | 12.9 | 13.5 | 11.5 | 8.36 | 36.7 |
| FA profile, g/100 g fat | | | | | | | | |
| C16:0 | 19.8 | 7.1 | 17.0 | 15.8 | 13.0 | 35.7 | 33.7 | 37.8 |
| C18:0 | 3.5 | 3.2 | 4.7 | 4.4 | 2.6 | 5.3 | 4.2 | 33.4 |
| C18:1 | 19.6 | 10.5 | 15.0 | 19.4 | 33.9 | 39.3 | 38.1 | 23.4 |
| C18:2 | 48.8 | 20.0 | 40.0 | 52.9 | 49.5 | 19.7 | 19.7 | 5.3 |
| C18:3 | 8.35 | 59.0 | 27.0 | 7.5 | 0.9 | 0.0 | 0.0 | 0.0 |
| SFA | 23.3 | 10.3 | 21.7 | 20.2 | 15.6 | 41.0 | 37.9 | 71.2 |
| MUFA | 57.2 | 79.0 | 67.0 | 60.4 | 50.4 | 19.7 | 19.7 | 5.3 |
| PUFA | 2.5 | 7.7 | 3.1 | 3.0 | 3.2 | 0.5 | 0.5 | 0.1 |

†Expressed as g/kg of fresh matter.

SBM, soybean meal.

as antioxidants, and two diets with 61 g/ kg DM of either chia seeds (CS) or pumpkin seeds (PS) as sources of protein and fat (Table 2).

Control diet, CS, and PS diets were supplemented with butter (13 g/kg DM), chia seed (61 g/kg DM), and pumpkin seed (61 g/kg DM), respectively. Contingent to the treatment, the concentrate was offered at 50 g/kg BW$^{0.75}$, and corn silage was fed *ad libitum*. The concentrate was manually mixed with the ingredients of each diet in batches of 100 kg as fresh matter.

**Table 2. Ingredients and nutrient composition of diets (DM basis) in lactating dairy ewes.**

| | Diets[2] | | |
|---|---|---|---|
| | **CON** | **PS** | **CS** |
| Ingredients | | | |
| Corn silage, g / kg | 430 | 430 | 430 |
| Chia seeds, g / kg | 0 | 0 | 61 |
| Pumpkin seeds, g / kg | 0 | 61 | 0 |
| Soybean meal, g / kg | 140 | 105 | 105 |
| Sorghum grain, g / kg | 250 | 237 | 237 |
| Oats hay, g / kg | 88 | 88 | 88 |
| Wheat bran, g / kg | 48 | 48 | 48 |
| Vitamins and minerals[1], g / kg | 31 | 31 | 31 |
| Butter INCA®, g / kg | 13 | 0 | 0 |
| Chemical composition | | | |
| Dry matter, g / kg | 967 | 972 | 968 |
| Organic matter, g / kg | 910 | 924 | 922 |
| Crude protein, g / kg | 143 | 142 | 142 |
| Ether extract, g / kg | 57 | 55 | 57 |
| Non fiber carbohydrates, g / kg | 428 | 424 | 410 |
| Neutral detergent fiber, g / kg | 281 | 303 | 313 |
| Acid detergent fiber, g / kg | 163 | 177 | 179 |
| Acid detergent lignin, g / kg | 21.0 | 37.9 | 52.7 |
| Ca, g / kg | 7.4 | 7.2 | 7.6 |
| P, g / kg | 5.2 | 5.1 | 5.6 |
| Metabolizable energy, MJ /kg DM | 11.6 | 11.5 | 11.5 |
| Fatty acids profile | | | |
| C16:0, g/100 g fat | 19.8 | 19.0 | 18.4 |
| C18:0, g/100 g fat | 3.6 | 3.5 | 3.4 |
| C18:1, g/100 g fat | 24.8 | 24.6 | 24.4 |
| C18:2, g/100 g fat | 43.5 | 43.4 | 42.2 |
| C18:3, g/100 g fat | 4.9 | 6.2 | 8.2 |
| SFA, g/100 g fat | 25.1 | 25.6 | 24.9 |
| MUFA, g/100 g fat | 24.8 | 24.6 | 24.4 |
| PUFA, g/100 g fat | 48.3 | 49.7 | 50.4 |

[1]Containing calcium (4.5 g/kg); cobalt (0.090 g/kg); copper (6 g/kg); ethylene-dynamine (0.500 g/kg); iron (20 g/kg); ionophore (30 g/kg); magnesium (500 mg/kg); manganese oxide (36 g/kg); potassium chloride (140 g/kg); salt (6 g/kg); selenium (0.090 g/kg); sodium (125 g/kg); vitamin A (3,000,000 IU/kg); vitamin D (3,700,000 IU/kg); vitamin E (18,000 IU/kg); antioxidant (25 mg/kg); zinc (50 g/kg).

[2]CON, PS, and CS diets were supplemented with INCA® butter, pumpkin seed, and chia seed, respectively, as FA sources.

## Measurements and laboratory analysis

The quantities of diet served, and orts were weighed daily for each ewe, however, only data of the last 5 days were considered for analysis. Daily feces and urine samples (using 10% sulphuric acid to maintain a pH < 3.0) were also taken on the last 5 days and then were frozen at −20˚C. On the last 7 days of the trial, fecal sampling was done at two-time points (6 h-prior and 6 h-post feeding). Dried fecal samples (at 60˚C for 72 h) were ground (using a 1 mm sieve) and then mixed in equal parts to obtain one fecal sample per ewe and preserved at -20˚C until further analysis.

The samples of ingredients, diets, refusals, and feces were dried at 60 ˚C for 48 h, pooled and ground in a hammer mill (Arthur Hill Thomas Co., Philadelphia, Pennsylvania, USA) then analyzed as described by the Association of Official Analytical Chemists (AOAC) [15] for ash (967.05), crude protein (CP; 990.03), ether extract (EE; 945.16), and organic matter (OM, 942.05). The contents of ash-free neutral detergent fiber (NDF), acid detergent fiber (ADF), and lignin were measured according to Van Soest [16]. All chemical analyses were performed in triplicate. The mineral content (Ca, and P) was determined according to Plank [17]. Fatty acids were measured following the method of Palmquist and Jenkins [18]. At the beginning and the end of each period, all animals were weighed following a 16-h fasting using a calibrated scale, and then individual body weight changes (BWC, g/d) were determined.

Individual milk samples (100 mL) with preservative (potassium bichromate) were collected on the last 5 days of the trial (at 16:00 h). Total solids (TS) and non-fat solids (NFS) were analyzed using a MilkoScan analyzer (SL60, Milkotronic LTD, Nova Zagora, Bulgaria). Milk protein and milk fat were measured as outlined by McKenzie and Murphy [19] and Levowitz [20], respectively. Milk urea nitrogen (MUN) was assessed using the micro-Kjeldahl method [15]. Fat-corrected milk (FCM) and fat-protein-corrected milk (FPCM) were calculated according to Pulina et al. [21]. Feed efficiency [FE; milk yield (kg/d) / dry matter intake, DMI (kg/d)] and adjusted FE [6.5% FCM (kg/d) / DMI (kg/d)] were also calculated.

## Milk Fatty Acid (FA) profile

The milk FA composition was determined following Frank et al. [22]. Briefly, a mixture of 250 mL milk sample and 250 mL of detergent solution [50 g of sodium hexametaphosphate plus 24 mL of Triton X100 in one liter of distilled water] was heated in a water bath at 90 ºC ± 2 ºC. The separated fat content was then filtered through number 4 Whatman paper with anhydrous sodium sulfate. Extracted was stored at -20 ºC for analysis. The fatty acids methyl esters (FAME) preparation followed ISO-IDF [23] guidelines using a gas chromatograph (Shimadzu GC-2010 Plus) equipped with CP-SIL-88 capillary column (100 m × 0.25 mm, Varian) and nitrogen was the gas carrier. The column temperature was initially 140ºC, increasing by 5ºC/min to 195ºC. The temperature of the injector was 250 ˚C, and that of the detector was 270 ˚C. The total running time was 50.17 minutes. To calculate the response factor of individual FA, butterfat (reference material CRM 164, European Community Bureau of Reference, Brussels, Belgium) was utilized [24]. Additionally, a standard consisting of 37 components (Supelco No. Cat. 47885-U. 33) was employed for the identification of the FA [25,26].

## In-vitro gas production (IVGP)

The IVGP was conducted by the protocol outlined by Theodorou et al. [26]. In short, dietary treatments (0.800 g DM) were placed in glass vials (125 mL) in triplicate, with four incubation runs (i.e., a total of 12 replicates per diet). To each bottle, rumen fluid (10 mL) and buffer solution (90 mL; the composition of which has been previously reported [27]) were added. The ruminal fluid used in the experiment was collected from three adult sheep (42 ± 2 kg of live

weight) that were fed CON diet. The fluid was subsequently filtered through triple cheesecloth gauze, homogenized with $CO_2$, and then added to the bottles. The bottles were placed in an oven for incubation at 39 ˚C and the produced gas was recorded at regular intervals of 3, 6, 9, 12, 24, 36, 48, 60, 72, 84, and 96 h. Upon completion of the 96-hour incubation period, total gas yield, relative gas yield, pH, and dry matter disappearance (DMD) were assessed [28,29].

## Statistical analysis

The IVGP data were analyzed using the general linear model (GLM) procedure of SAS version 9.2, while *in-vivo* data was analyzed based on a Latin square design. The model included the fixed effects of treatments, experimental period, and the random effect was the sheep. Tukey's test was utilized to calculate the least-squares means (LSM) and determine any significant differences. A significance level of $p \leq 0.05$ was applied for all data.

## Results

Data sets from this study are found as (S1 File).

### Intake and digestibility

Final live weight (LW, $p > 0.05$), intake of DM ($p > 0.05$), and OM ($p > 0.05$) were not altered by diets (Table 3). On the contrary, intake of NDF ($p < 0.001$), ADF ($p < 0.01$), and ADL

**Table 3. Effect of dietary butter (CON), chia seed (CS) and pumpkin seed (PS) on nutrients intake and digestibility in dairy sheep.**

| Parameters | Diets | | | SEM | P-value |
|---|---|---|---|---|---|
| | CON | PS | CS | | |
| Final live weight, kg | 78.7 | 79.07 | 78.51 | 1.85 | 0.978 |
| Metabolic live weigh, $LW^{0.75}$ | 26.3 | 26.46 | 26.28 | 0.47 | 0.965 |
| Intake, g $d^{-1}$ | | | | | |
| Concentrate | 1386 | 1404 | 1420 | 16.8 | 0.376 |
| Forage | 1078 | 1104 | 1091 | 52.5 | 0.941 |
| Dry matter | 2464 | 2508 | 2512 | 56.4 | 0.806 |
| Organic matter | 2326 | 2368 | 2370 | 55.8 | 0.822 |
| Neutral detergent fiber | 1061[b] | 1171[b] | 1402[a] | 36.3 | 0.001 |
| Acid detergent fiber | 680[b] | 746[ab] | 800[a] | 25.7 | 0.006 |
| Acid detergent lignin | 110[c] | 136[b] | 157[a] | 3.99 | 0.001 |
| Ether extract | 188[b] | 204[a] | 197[ab] | 3.50 | 0.006 |
| Intake, g/kg $LW^{0.75}$ $d^{-1}$ | | | | | |
| Dry matter | 93.1 | 95.6 | 97.1 | 2.23 | 0.454 |
| Organic matter | 87.8 | 90.3 | 91.5 | 2.17 | 0.480 |
| Neutral detergent fiber | 39.9[c] | 44.7[b] | 54.1[a] | 1.39 | 0.001 |
| Acid detergent fiber | 25.5[b] | 28.4[ab] | 30.8[a] | 0.95 | 0.001 |
| Acid detergent lignin | 4.16[c] | 5.21[b] | 6.07[a] | 0.15 | 0.001 |
| Ether extract | 7.14[b] | 7.79[a] | 7.65[ab] | 0.15 | 0.007 |
| Digestibility coefficient, g/kg | | | | | |
| Dry matter | 710 | 710 | 710 | 11.1 | 0.841 |
| Organic matter | 727 | 719 | 724 | 10.0 | 0.844 |
| Neutral detergent fiber | 612[b] | 628[b] | 701[a] | 14.0 | 0.001 |

[a-c] Within a row, means with different superscript letters are different ($p \leq 0.05$).

SEM = standard error the mean.

**Table 4. Effect of dietary butter (CON), chia seed (CS) and pumpkin seed (PS) on nitrogen balance in dairy sheep.**

| Parameters | Diets | | | SEM | P-value |
|---|---|---|---|---|---|
| | CON | PS | CS | | |
| Nitrogen balance | | | | | |
| N intake g/d | 57.40[b] | 71.30[a] | 71.33[a] | 0.938 | 0.001 |
| N intake g/d, LW[75] | 2.18[b] | 2.71[a] | 2.76[a] | 0.049 | 0.001 |
| N excreted, d[-1] | | | | | |
| Urinary N | 33.83[b] | 45.91[a] | 45.90[a] | 0.480 | 0.001 |
| Fecal N | 20.43 | 20.04 | 19.34 | 0.908 | 0.704 |
| Milk N | 5.95[b] | 5.97[b] | 6.38[a] | 0.086 | 0.001 |
| N Balance g d[-1] | -2.81[b] | -0.62[a] | -0.31[a] | 0.832 | 0.037 |
| Fecal N / N Intake % | 35.72[a] | 28.04[b] | 27.02[b] | 1.309 | 0.001 |
| Urinary N/ N Intake % | 59.05[b] | 64.48[a] | 64.45[a] | 0.298 | 0.001 |
| Milk N / N Intake % | 10.62[a] | 8.41[b] | 8.97[b] | 0.213 | 0.001 |

[a,b] Within a row, means with different superscript letters are different ($p \leq 0.05$).

SEM, standard error the mean.

($p < 0.001$) were increased with either CS and/or PS. Also, the highest intake of EE ($p < 0.01$) occurred in sheep fed PS. Dietary inclusion of CS and PS did not alter the digestibility of DM ($p > 0.5$) and OM ($p > 0.5$) while increased digestibility of NDF ($p < 0.001$).

## Nitrogen balance

The nitrogen (N) balance results are given in Table 4. Both CS and PS led to a significant enhancement ($p < 0.001$) in N uptake. Moreover, the highest milk N retention ($p < 0.05$) was observed in sheep fed CS followed by PS.

## Milk yield and milk components

Table 5 presents the data on lactating performance and milk components of dairy sheep. Dietary inclusion of PS results in higher Milk yield ($p < 0.001$), 6.5% FCM ($p < 0.01$), and MUN ($p < 0.001$). Indeed, milk composition (g/100g) including protein ($p < 0.01$), lactose ($p < 0.001$), and total solids ($p < 0.05$) were affected by diets (Table 5). Also, CS supplementation increased yields (g/d) of protein ($p < 0.01$), lactose ($p < 0.001$), and total solids ($p < 0.05$). The inclusion of PS yielded the highest fat content, whereas the CS was accompanied by the lowest fat content ($p < 0.001$).

## Milk fatty acid profile

The milk FA profile of ewes is shown in Table 6. The concentrations of short-chain FA (SCFA; C4-C8) in milk were not altered by diets ($p > 0.05$). The ingestion of either CS or PS decreased the C16:0 in milk fat ($p < 0.001$). Compared to the CON, supplementation of CS to dairy ewes enhanced C18:0 in milk fat ($p < 0.001$), too C18:1n9 (oleic acid) increased ($p < 0.01$). The contents of C18:2n6 trans (linolelaidic acid; $p > 0.05$) and C18:2n6 (linoleic acid; $p > 0.05$) were not affected by oilseeds. The content of C20:0 (arachidic acid) enhanced ($p < 0.001$) in CS-supplemented ewes compared with CON and PS. In addition, both CS and PS tended to increase the content of C18:3n3 ($\alpha$-linolenic acid; $p > 0.05$). Although the concentration of SFA decreased ($p < 0.01$), MUFA increased ($p < 0.001$), and PUFA remained unaffected ($p > 0.05$) with the inclusion of CS and PS.

**Table 5. Effect of dietary butter (CON), chia seed (CS) and pumpkin seed (PS) on milk yield and composition in dairy sheep.**

| Parameters | Diets | | | SEM | P-value |
|---|---|---|---|---|---|
| | CON | PS | CS | | |
| Milk yield, kg/d | 0.37[a] | 0.34[a] | 0.21[b] | 29.4 | 0.001 |
| Milk DMI, kg /d | 0.15[a] | 0.13[a] | 0.08[b] | 0.01 | 0.002 |
| Fat-corrected milk, DMI | 0.14[a] | 0.12[a] | 0.07[b] | 0.01 | 0.001 |
| Feed efficiency FCM, DMI | 0.15[a] | 0.13[a] | 0.08[b] | 0.01 | 0.001 |
| Fat-corrected milk 6.5% | 0.34[a] | 0.34[a] | 0.19[b] | 0.02 | 0.008 |
| Fat protein- corrected | 0.34[a] | 0.33[a] | 0.19[b] | 0.02 | 0.010 |
| Milk-N/N-Intake % | 10.6[a] | 8.41[b] | 8.97[b] | 0.21 | 0.001 |
| MUN, mg/dl | 10.7[b] | 16.2[a] | 16.2[a] | 0.30 | 0.001 |
| Milk composition g/100g | | | | | |
| Fat | 5.89 | 6.05 | 5.42 | 0.15 | 0.132 |
| Protein | 5.04[b] | 5.05[b] | 5.39[a] | 0.06 | 0.016 |
| Lactose | 4.77[b] | 4.89[ab] | 5.09[a] | 0.06 | 0.001 |
| Total solids | 10.6[b] | 10.7[b] | 11.3[a] | 0.14 | 0.037 |
| Milk composition, g/d | | | | | |
| Fat | 44.5[b] | 46.0[a] | 41.0[c] | 1.15 | 0.001 |
| Protein | 38.0[b] | 38.3[b] | 40.9[a] | 0.52 | 0.016 |
| Lactose | 36.0[b] | 37.0[ab] | 38.3[a] | 0.51 | 0.001 |
| Total solids | 80.2[b] | 80.3[b] | 85.4[a] | 1.07 | 0.037 |

[a,b] Within row, means with different superscript letters are different ($p \leq 0.05$).

SEM, standard error the mean.

## In vitro gas production

Fractional rate of degradation (c; $p > 0.05$) and lag time ($p > 0.05$) remained unaffected by diets (Table 7). Gas production at 6h ($p < 0.01$) and 9h ($p < 0.01$) was greater for PS diet. In addition, compared to the control, diets with seeds (CS and/or PS) decreased the disappearance of DM at 96 h (DMD96; $p < 0.001$) as well as MCP ($p < 0.001$).

## Discussion

### Nutrient intake and digestibility

Our results showed that the intakes of NDF, ADF, and ADL were increased with CS and PS. While previous studies have suggested that the inclusion of CS and PS may enhance the acceptability of concentrates to ruminants [4,30], in this study, the intake of both DM and OM were similar. However, our results are consistent with those found by Schettino et al. [31] in dairy goats fed different levels of CS and those reported by Li et al. [4] when replacing SBM with PS in dairy cow diets. Moreover, similar findings were documented by Cardoso-Gutiérrez et al. [3] using sunflower seeds in dairy sheep diets. It has been well documented that the use of oil seeds helps to maintain body condition in sheep since they are an efficient energy source [8]. In addition, Schettino et al. [31] reported higher NDF and ADF intake following supplementation of CS at 5.5% DM in dairy goats. Also, our results showed that dietary inclusion of PS was accompanied by the highest intake of ether extract and that may be associated with the high contents of fat in PS as reported by Li et al. [4].

Also, the current results showed that the digestibility of NDF was enhanced with the inclusion of CS and PS. Although the roughage composition of the three diets used in this study

**Table 6. Effect of dietary butter (CON), chia seed (CS) and pumpkin seed (PS) on FA profile (g /100 g FA total methyl ester) in sheep milk.**

| FA | Diets | | | | |
|---|---|---|---|---|---|
| | CON | PS | CS | SEM | *P-value* |
| C4:0 | 3.50 | 3.77 | 3.42 | 0.23 | 0.397 |
| C6:0 | 2.51 | 2.73 | 2.41 | 0.14 | 0.459 |
| C8:0 | 2.29 | 2.50 | 2.16 | 0.13 | 0.155 |
| C10:0 | 8.62[a] | 8.90[a] | 7.60[b] | 0.22 | 0.003 |
| C11:0 | 0.44[a] | 0.47[a] | 0.32[b] | 0.02 | 0.001 |
| C12:0 | 4.61[a] | 4.61[a] | 3.89[b] | 0.09 | 0.001 |
| C14:0 | 12.9[a] | 12.2[b] | 12.0[b] | 0.13 | 0.001 |
| C14:1 | 0.38 | 0.32 | 0.34 | 0.02 | 0.523 |
| C15:0 | 1.31[a] | 1.19[a] | 0.90[b] | 0.07 | 0.005 |
| C15:1 | 0.18 | 0.16 | 0.38 | 0.06 | 0.103 |
| C16:0 | 32.4[a] | 28.8[b] | 29.9[b] | 0.42 | 0.001 |
| C16:1 | 1.43 | 1.42 | 1.40 | 0.11 | 0.986 |
| C17:0 | 0.68[a] | 0.55[b] | 0.50[b] | 0.02 | 0.001 |
| C17:1 | 0.41[a] | 0.29[b] | 0.31[b] | 0.00 | 0.001 |
| C18:0 | 7.15[c] | 8.55[b] | 10.2[a] | 0.13 | 0.001 |
| C18:1t | 2.20[a] | 1.41[b] | 1.08[b] | 0.20 | 0.010 |
| C18:1 9c | 16.0[b] | 19.4[a] | 20.4[a] | 0.37 | 0.001 |
| C18:2n6t | 0.19 | 0.23 | 0.24 | 0.01 | 0.144 |
| C18:2n6c | 1.70 | 1.74 | 1.56 | 0.06 | 0.247 |
| C20:0 | 0.10[b] | 0.10[b] | 0.13[a] | 0.002 | 0.001 |
| C18:3n6 | 0.07 | 0.07 | 0.05 | 0.004 | 0.062 |
| C18:3n3 | 0.09 | 0.20 | 0.21 | 0.03 | 0.081 |
| C18:2cis9trans11 | 0.40 | 0.37 | 0.36 | 0.02 | 0.553 |
| C20:3n3 | 0.15 | 0.17 | 0.14 | 0.008 | 0.266 |
| ΣSFA | 76.6[a] | 74.4[b] | 73.6[b] | 0.50 | 0.002 |
| ΣMUFA | 20.6[b] | 23.0[a] | 23.9[a] | 0.38 | 0.001 |
| ΣPUFA | 2.61 | 2.76 | 2.58 | 0.07 | 0.262 |

[a-c] Within a row, means with different superscript letters are different ($p \leq 0.05$).

SEM, standard error the mean.

was identical as well as similar protein and energy contents, the digestibility of NDF was enhanced with oil seeds, and the presence of high content of PUFA in the seeds may be responsible for these effects. Additionally, high PUFA could produce defaunation of ciliate protozoa predators in the rumen, which could increase the populations of cellulolytic bacteria, and thereby improve NDF digestibility [32]. Conversely, Schettino et al. [31] found that the nutrient digestibility of goats fed with increasing levels of CS remained unchanged.

### Nitrogen balance

Our results showed that the N uptake was higher with both CS and PS. Also, the highest N balance was observed in CS followed by PS. Moreover, the amounts of Fecal N (20.0 ± 0.7 g/d) were lower than those in urinary (45.9 ± 0.01 g/d), suggesting that there was greater use of ruminal ammonia [33]. It has been mentioned that N losses are from 70 to 95% and occur mainly through urine [34], which coincides with our current findings. In the present study, positive balance was obtained for the three treatments; however, it was higher for the ewes that

**Table 7. Effect of butter (CON), chia seed (CS) and pumpkin seed (PS) on *in vitro* gas production.**

| Parameters | Diets | | | SEM | P-value |
|---|---|---|---|---|---|
| | CON | PS | CS | | |
| A | 204 | 242 | 238 | 11.1 | 0.059 |
| B | 0.052[a] | 0.047[b] | 0.045[b] | 0.001 | 0.001 |
| C | -0.027 | -0.025 | -0.022 | 0.01 | 0.957 |
| Lag time | 1.01 | 0.75 | 0.73 | 0.17 | 0.449 |
| Gas production, mL gas/g DM | | | | | |
| 6 h | 32.3[b] | 39.3[a] | 37.0[ab] | 1.40 | 0.009 |
| 9 h | 62.0[b] | 70.3[a] | 69.0[ab] | 1.93 | 0.017 |
| 12 h | 95.0 | 102 | 101 | 4.19 | 0.126 |
| 24 h | 133 | 151 | 149 | 6.36 | 0.116 |
| 48 h | 179 | 207 | 201 | 9.18 | 0.107 |
| 96 h | 207 | 242 | 239 | 10.7 | 0.069 |
| DMD 96h | 80.6[a] | 78.0[b] | 76.3[c] | 0.27 | 0.001 |
| PF 96h | 257.0[b] | 310[a] | 313[a] | 13.3 | 0.015 |
| GY 24 h | 26.6[b] | 30.3[a] | 30.0[a] | 1.29 | 0.024 |
| SCFA | 22.0[b] | 33.0[a] | 19.0[c] | 0.01 | 0.001 |
| MCP | 715[a] | 675[b] | 658[b] | 5.07 | 0.001 |

[a -c] Within a row, means with different superscript letters are different ($p \leq 0.05$);

SEM, standard error the mean; A, total gas production; B, fermentation rate; C, fermentation rate; DMD, DM degraded substrate; GY, gas yield; SCFA, short chain fatty acids; MCP, microbial CP production.

consumed CS, likely due to the increased N intake, which may reflect lower mobility of body reserves [4,33]. However, there were no differences in live weight and milk production. Moreover, the animals fed the either PS or CS diets, having numerically higher DMI (+1.8% and +1.0%, respectively) in comparison with the CON diet but the same milk yield showed a higher N balance (g/d). Data from this study shows the capacity of CS or even PS to supply the required N for dairy sheep while satisfying the protein needs of the rumen microbiome [8], and the amount of oil present in the seeds could favor ammonia generated via the degradation of proteins in the rumen without affecting cellulolytic bacteria [27]. Additionally, the improvement in N balance with both CS and/or PS may be related to the reduction in intestinal viscosity promoted by the inclusion of oil in the diet since this effect can lead to improved digestion and nutrient absorption in the small intestine [3].

## Milk yield and milk components

Milk yield and FCM were higher in PS compared to CS. Commonly, a higher milk yield is expected via concentrate-rich diets, however, this was not the case for CS, although the forage: concentrate ratio was approximately 44:56 in the CON diet as in CS, the higher amount of fiber fractions (NDF, ADF, and lignin) and a lower content of non-structural carbohydrates (NSC), such as sugars and starch could be the result of the lower amount of milk obtained [35] reflected in our study. In addition, the present results showed that the CS supplementation increased yields (g/d) of lactose, total solids, and protein which did not occur in goats fed different inclusion levels of CS [31] or in goats fed a grass silage-based diet supplemented with whole sunflower or flaxseed [27]. Likewise, contents of protein, lactose, and fat remained unaffected in goats fed PS [8].

It has been observed that the inclusion of fat in ruminant diets reduces the fermentable organic matter (OM), glucose precursors, and microbial protein synthesis in the rumen, thereby affecting the reservoir of amino acids (AAs) ready for protein synthesis in milk [36]. Bartocci et al. [37] showed that in lactating goats, milk protein percentage was not affected by cottonseed inclusion (up to 18% DM of diet). An elevation in lactose concentration following the ingestion of crushed flaxseed and flaxseed oil was reported by Kholif et al. [30], which may be attributed to the higher propionate production that serves as a precursor for lactose synthesis and gluconeogenesis.

Milk fat content was enhanced with the use of PS (46.0 g/d), however with CS the opposite occurred (41.0 g/d), being the treatment with the lowest amount of fat. The type and amount of PUFA present in CS could induce a milk fat depression (MFD) [30,38].

A study involving sheep and goats did not establish a significant relationship between the intensity of MFD and the content of antilipogenic C18 FA formed in the rumen, such as trans-10 18:1, trans-10, cis-12 CLA, or trans-9, cis-11 CLA [39]. Della Badia et al. [38] concluded that based on the commonality of the responses in ruminant species goat and sheep, the tolerance or susceptibility to MFD may depend predominantly on individual differences in the extent of BH of certain potentially antilipogenic UFA provided by fish oil. The diet with PS could provide enough effective fiber which helped to increase milk fat [3,25].

## Milk fatty acid profile

Some studies have found that the milk FA profile can be regulated by changing the dietary composition or supplementing additional oilseeds [2,40]. The composition of milk FA is dependent on two key factors: rumen metabolism (involving hydrolysis, isomerization, and biohydrogenation of dietary FA, which ultimately determine the flow and composition of FA in the duodenum) and animal metabolism, on the other hand (encompasses lipid mobilization and mammary uptake and synthesis of FA). Together, these processes integrate to determine the overall response of the milk FA profile [24]. In the current investigation, the content of SCFA remained unchanged. Similar results were obtained in goat milk when a diet enriched with flaxseed and chia oil [30,40] and whole linseed and sunflower oil [25,41] were fed, which could be explained by their synthesis in the mammary from ruminal β-hydroxybutyrate [18]. Dairy lipids draw attention due to their impact on human health, given that myristic acid (C14:0) and palmitic acid (C16:0) have been linked to elevated blood cholesterol levels, potentially posing a risk for cardiovascular diseases [1,4,42]. Both CS and PS decreased the concentration of C14-C16 FA in sheep milk. Bernard et al. [41] reported that the rise in long-chain FA (LCFA) accumulated by the mammary gland led to a decline in the functionality of enzymes participating in the synthesis of FA. Della Badia et al. [39] found a negative relationship between high ingestion of unsaturated plant lipids and the chain length of saturated milk FA. Specifically, longer-chain saturated FAs are more greatly impacted by the ingestion of unsaturated lipids derived from plant sources.

Both CS and PS led to higher content of C18:0 in ewe milk, which can be attributed to the full biohydrogenation of the linolenic acid and linoleic acid and in the rumen to C18:0 since the delivery of lipids is slower with oilseeds than with pure oils [25,38]. In confirmation, Chilliard et al. [25] speculated that administration of un-protected oilseed primarily leads to an elevation in the content of C18:0 and C18:1 in milk which are likely attributable to the alterations in the pathways involved in ruminal biohydrogenation of dietary C18:2 cis-9 cis-12. Linolenic acid (C18:3) in milk originates almost entirely from the diet. The content of C18:3n-3 in milk of sheep fed either CS and/or PS was greater than from sheep fed the CON diet. These findings are consistent with the reports of Ashes et al. [43]. It has been well

established that most C18:1 cis-9, C18:2 n-6, and C18:2 n-3 in the CS and PS are hydrogenated by microorganisms after entering the rumen, which will produce C18 and various isomers of MUFA and PUFA [2,44]. These products are absorbed by the intestinal tract and used for various purposes; some are directly passed into the milk, and some are converted by the mammary gland tissue [25].

Similar to our results, the inclusion of PS in Alpine goats [45] and Holstein cows [2] during early lactation had little effect on the FA content in milk being higher content of C18:0 along with lower SCFA, proving the complexity of regulating FAs, due in part to the complexity of rumen microbial responses [4,38]. Moreover, Schettino et al. [31] by using 2.7 or 5.5% (DM basis) CS for dairy goats, noted that there was no improvement in milk yield, however, milk FA profile was altered, and there was a reduction in the proportion of medium-chain FA (MCFA) (i.e., C12:0-C16:0). In the same study, MUFA and PUFA (like C18:1n-9 cis and C18:2 cis-9 trans-11) enhanced as compared with the control. The FA in milk comes from *de novo* synthesis in the mammary gland (mainly short- and medium-chain SFA) or from plasma FA absorbed through the ruminal wall, that is, LCFA and MUFA [25]. Our results showed that the SFA decreased with CS and PS, whereas MUFA increased, suggesting that the elevation in the dietary lipid supply was linked to a decline in *de novo* formation of FA [41]. It is also well established that the dietary inclusion of oilseed led to lower concentrations of SFA, in line with a decline in mammary FA synthesis which is offset by an elevation in the content of C18 FA in milk [25,39]. Moreover, this decrease may be because they are mainly synthesized *de novo* in the mammary gland which could be impeded by the trans FA formed from the biohydrogenation of PUFA in the rumen [27,31]. In confirmation, Schettino et al. [31] reported a higher MUFA in goats fed CS, which researchers linked to the partially biohydrogenation of FA such as linoleic acid in the rumen as well as the action of the $\Delta^9$desaturase on stearic acid.

It is notorious that our control treatment was made from hydrogenated vegetable and animal oils and even at 1.5% DM supplementation, this was enough to affect milk FA, and that reflected in increased contents of lauric acid, myristic acid, palmitic acid, and total SFA which are considered as negative for human health. Saturated FA have been linked to a higher incidence of cardiometabolic diseases [44]. Therefore, from a human standpoint, compared to CS and PS, using saturated FA sources seems to be detrimental to milk FA profile.

## In vitro gas production

Gas production is commonly considered an indirect indicator of substrate degradation, particularly those derived from carbohydrates. Additionally, it serves as a reliable predictor of microbial crude protein (MCP) and SCFA production [29]. The present results revealed that the gas production at 6h and 9h was higher for the PS diet, however, fermentation rate and lag time remained unaffected. In addition, both CS and PS decreased the DMD96 as well as MCP. The oil contained in the seeds may have had a disruptive effect on the rumen microbial ecosystem, thereby inhibiting microbial activity, specifically those of cellulose-fermenting and methane-producing microorganisms, resulting in a decrease in overall microbial fermentation [27]. The above contrasts with what was indicated by Silva et al. [46] and Schettino et al. [31] mentioning that CS in *in-vitro* cultures increased fluxes of α-linolenic acid, arachidonic acid, and total PUFA indicating that CS were extensively biohydrogenated in the rumen. However, these results conflict with the *in-vivo* data of the current investigation, since NDF digestibility was higher in the diets with CS and PS, MCP production was probably also improved, as we can observe in Table 4 for the N retention. The increased presence of NDF in seed diets may have led to a reduction of the suppressive effects of vegetable oil on cellulolytic bacteria, contributing to a higher efficiency of dietary protein utilization [3,32].

## Conclusion

Results showed that dietary inclusion of 6.1% DM of CS and/or PS reduced C14:0 and C16:0 in sheep milk while increasing C 20:0 and total contents of MUFA with a tendency for C18:3n3 suggesting that the incorporation of both oil seeds modifies the milk FA to be more healthful for human consumption, without affecting animal performance. Overall short-term feeding of CS and/or PS (up to 6.1% DM of diet) improves milk FA profile without deleterious effects on performance and digestibility of nutrients.

## Supporting information

**S1 File. "Data sets from this study".**
(XLSX)

## Acknowledgments

The authors express their gratitude to the staff of the Universidad Autónoma del Estado de México and Universidad Juárez Autónoma de Tabasco for their contribution to the present study and animal care. Miss. Lizbeth E. Robles Jimenez (LERJ) was granted with a CONACyT, Estancias postdoctorales 2021.

## Author Contributions

**Conceptualization:** Lizbeth E. Robles Jimenez, Edgar Aranda Aguirre, Maria de los Angeles Colin Cruz, Alfonso J. Chay-Canul, Ricardo A. Garcia-Herrera, Manuel Gonzalez-Ronquillo.

**Data curation:** Lizbeth E. Robles Jimenez, Alfonso J. Chay-Canul, Einar Vargas-Bello-Pérez, Manuel Gonzalez-Ronquillo.

**Formal analysis:** Lizbeth E. Robles Jimenez, Maria de los Angeles Colin Cruz, Beatriz Schettino-Bermúdez, Alfonso J. Chay-Canul, Manuel Gonzalez-Ronquillo.

**Funding acquisition:** Manuel Gonzalez-Ronquillo.

**Investigation:** Lizbeth E. Robles Jimenez, Alfonso J. Chay-Canul, Einar Vargas-Bello-Pérez, Manuel Gonzalez-Ronquillo.

**Methodology:** Maria de los Angeles Colin Cruz, Alfonso J. Chay-Canul, Ricardo A. Garcia-Herrera, Octavio A. Castelan Ortega, Einar Vargas-Bello-Pérez, Manuel Gonzalez-Ronquillo.

**Project administration:** Manuel Gonzalez-Ronquillo.

**Resources:** Manuel Gonzalez-Ronquillo.

**Software:** Maria de los Angeles Colin Cruz, Ricardo A. Garcia-Herrera, Octavio A. Castelan Ortega, Manuel Gonzalez-Ronquillo.

**Supervision:** Lizbeth E. Robles Jimenez, Octavio A. Castelan Ortega, Einar Vargas-Bello-Pérez, Manuel Gonzalez-Ronquillo.

**Validation:** Lizbeth E. Robles Jimenez, Beatriz Schettino-Bermúdez, Rey Gutiérrez-Tolentino, Octavio A. Castelan Ortega, Manuel Gonzalez-Ronquillo.

**Visualization:** Lizbeth E. Robles Jimenez, Rey Gutiérrez-Tolentino, Octavio A. Castelan Ortega, Manuel Gonzalez-Ronquillo.

**Writing – original draft:** Lizbeth E. Robles Jimenez, Beatriz Schettino-Bermúdez, Rey Gutiér-rez-Tolentino, Alfonso J. Chay-Canul, Ricardo A. Garcia-Herrera, Navid Ghavipanje, Einar Vargas-Bello-Pérez, Manuel Gonzalez-Ronquillo.

**Writing – review & editing:** Navid Ghavipanje, Manuel Gonzalez-Ronquillo.

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
