## [Decision Letter · Decision Letter 0]

7 Sep 2023

PONE-D-23-22942Inclusion of chia seeds (Salvia hispanica L.) and pumpkin seeds (Cucurbita moschata) in dairy sheep dietsPLOS ONE

Dear Dr. Vargas-Bello-Pérez,

Thank you for submitting your manuscript to PLOS ONE. After careful consideration, we feel that it has merit but does not fully meet PLOS ONE’s publication criteria as it currently stands. Therefore, we invite you to submit a revised version of the manuscript that addresses the points raised during the review process.

Dear AuthorsPlease also check the result section carefully, because in result section especially in digestibility some values are under/over estimated. Failing to address the issues or reviewers comments will result in rejection of manuscript.

We look forward to receiving your revised manuscript.

Kind regards,

Aziz ur Rahman Muhammad

Academic Editor

PLOS ONE

Journal Requirements:

" The funders had no role in study design, data collection and analysis, decision to publish, or preparation of the manuscript "

Additional Editor Comments:

Dear Authors

Please revise the manuscript as suggested by reviewers. Please also check the result section carefully, because in result section especially in digestibility some values are under/over estimated.

Reviewers' comments:

Reviewer's Responses to Questions

**Comments to the Author**

1. Is the manuscript technically sound, and do the data support the conclusions?

Reviewer #1: Partly

Reviewer #2: Yes

Reviewer #3: Partly

2. Has the statistical analysis been performed appropriately and rigorously? 

Reviewer #1: Yes

Reviewer #2: Yes

Reviewer #3: N/A

3. Have the authors made all data underlying the findings in their manuscript fully available?

Reviewer #1: Yes

Reviewer #2: Yes

Reviewer #3: Yes

4. Is the manuscript presented in an intelligible fashion and written in standard English?

Reviewer #1: Yes

Reviewer #2: Yes

Reviewer #3: Yes

5. Review Comments to the Author

Reviewer #1: The paper exploring the inclusion of chia seeds (Salvia hispanica L.) and pumpkin seeds (Cucurbita moschata) in dairy sheep diets holds promise for enhancing the nutritional profile and overall health of the animals. Both chia and pumpkin seeds are rich sources of essential nutrients such as omega-3 fatty acids, protein, vitamins, and minerals. Incorporating these seeds into the diet could potentially lead to improved milk quality, enhanced immune function, and better reproductive outcomes for dairy sheep.

However, the paper may require corrections to address and establish a clear cause-and-effect relationship between seed inclusion and observed benefits. With these adjustments, the study could contribute valuable insights to optimizing dairy sheep nutrition and potentially benefit both animal health and the dairy industry.

The following major points should be addressed:

Line 70. I have to disagree with the authors. Not all pumpkin seeds contains that much SFA. The reference no (Yang Li et al., 2023) do not report the FA profile in the pumpkin seed cake, only in the feeds . In a recent study, it was reported that pumpkin seed meal contains up to 51% PUFA, among other nutrients (https://www.ncbi.nlm.nih.gov/pmc/articles/PMC8738952/), and because of that the above mentioned information do not stand. Please correct.

Line 95. … sample collection instead of harvesting.

Table 1. please provide at least for CS and PS the total PUFA and MUFA, the total n-3 and the total n-3, as well as their ratio. This is important and significant information.

Table 2. please add the measurement units.

Line 141. Please explain in words when using first time FAME (fatty acids methyl esters).

Table 3. the live weight, is initial or final?

Table 6. Please check the data for C18:3n3, no significant effect? Also for other fatty acids

In the discussion part, some parameters deserves more considerations to be discussed, especially Nitrogen balance.

The conclusion is not accurate. According to the data presented in table 6, the C18:3n3 is not significantly increased, only a tendency, this do not count as a major result.

Reviewer #2: The overall idea of this kind of studies is novel and worth's trying, it helps the livestock producers to find energy sources that may improve the quality of animals products such as milk. There are some issues must be addressed by the authors in order to improve the manuscript.

Line 33: delete "DM"

Line 34: please delete "LW"

Line 35: delete "DMI"

Line 36: delete "NDF"

Line 37: delete "ADL" and "FCM"

Line 38: delete "FE"

Line 41: when p value is 0.818, it is not considered tendency, please rewrite "Moreover,..... (p = 0.818).

Line 43: according to the results of FA composition in the milk, this statement is not true or at least not enough to draw this conclusion, please rewrite the conclusion statement.

Line 61-62: how much Mexico is producing from these two seed per year?

Line 67-68: Please add their nutrient content in values

- Table 2: please add the units of the ingredients and chemical composition. In the footnote, please define the diets

Line 162: don't start any statement with abbreviations

-Table 3: the digestibility of ADF and ADL *is not accurate please re-visit their actual values in check

- Table 4: the values of nitrogen balance are questionable, please re-check them. The N balance is too low

- Table 5: please recheck. for example milk composition (g/d) are all wrong

Although the discussion section is well written, it has to be rewritten to reflect the previous comments regarding some table (see previous comments)

Line 280 and conclusion: the positive results of FA composition in milk may not be related to CS and/or PS, it could be because the control group fed on diet contained butter! please clarify

Reviewer #3: When I came across the nutrient consumption table, I noticed a fact that will have to be reviewed for me to continue the review:

Let's take the CON treatment as an example:

In table 2 the proportion of corn silage was 350 g/kg

Table 3 describes:

Forage intake: 1078 g

Concentrate intake: 1386 g

DM intake: 2464g

Corn silage ratio: 1078/2464 = 456.14 g/kg (a value well above the 350 g/kg described in table 2)

NDF intake: 1061 g

DM intake: 2464g

NDF proportion in the diet effectively consumed:

1061/2464 = 456.14 g/kg, a value very, very different from that of table 2 (259 g/kg).

If we do the same thing for EE consumption, we will see that the diet effectively consumed showed values much higher than those described in Table 2.

Crude protein intake was not presented.

Digestibility coefficients draw a lot of attention:

Digestibility of ADF greater than NDF.

ADL digestibility of 430 g/kg, never seen before.

So, to follow the correction, I suggest that the calculations, spreadsheets, etc., be reviewed.

6. PLOS authors have the option to publish the peer review history of their article (what does this mean?). If published, this will include your full peer review and any attached files.

Reviewer #1: No

Reviewer #2: No

Reviewer #3: No

---

## [Author Response · Author response to Decision Letter 0]

27 Oct 2023

Cover letter 

[14th September 2023]

Dear Sirs,

On behalf of my coauthors, I would like to thank you for the opportunity to revise and resubmit our manuscript (ID: PONE-D-23-22942), entitled “Inclusion of chia seeds (Salvia hispanica L.) and pumpkin seeds (Cucurbita moschata) in dairy sheep diets”.

We found the reviewers’ comments to be helpful in revising the manuscript and have carefully considered and responded to each suggestion. We believe have resulted in an improved revised manuscript. Below we have included a response to each reviewer in which we address each comment the reviewers made. In our response to reviewers, the reviewers’ comments are numbered, and our responses follow below. Please be advised that we uploaded a marked-up copy of manuscript labeled 'Revised Manuscript with Track Changes' and an unmarked version of your revised paper without tracked changes labeled 'Manuscript'.

We very much hope the revised manuscript is accepted for publication in PLOS ONE.

Sincerely,

Einar Vargas-Bello-Perez (corresponding author)

School of Agriculture, Policy and Development

New Agriculture Building, Earley Gate

Whiteknights Road, PO Box 237

Reading RG6 6EU Berkshire UK

Response to Reviewer 1 Comments:

Reviewer #1: The paper exploring the inclusion of chia seeds (Salvia hispanica L.) and pumpkin seeds (Cucurbita moschata) in dairy sheep diets holds promise for enhancing the nutritional profile and overall health of the animals. Both chia and pumpkin seeds are rich sources of essential nutrients such as omega-3 fatty acids, protein, vitamins, and minerals. Incorporating these seeds into the diet could potentially lead to improved milk quality, enhanced immune function, and better reproductive outcomes for dairy sheep.

However, the paper may require corrections to address and establish a clear cause-and-effect relationship between seed inclusion and observed benefits. With these adjustments, the study could contribute valuable insights to optimizing dairy sheep nutrition and potentially benefit both animal health and the dairy industry. The following major points should be addressed:

Authors: Thank you for your valuable feedback on our manuscript. We appreciate your comments and suggestions, and we have carefully considered your feedback in preparing our revised manuscript.

Line 70: I have to disagree with the authors. Not all pumpkin seeds contains that much SFA. The reference no (Yang Li et al., 2023) do not report the FA profile in the pumpkin seed cake, only in the feeds. In a recent study, it was reported that pumpkin seed meal contains up to 51% PUFA, among other nutrients (https://www.ncbi.nlm.nih.gov/pmc/articles/PMC8738952/), and because of that the above mentioned information do not stand. Please correct

Authors: We acknowledge that there is conflicting results in literature for fatty acid composition of pumpkin seed oil which is probably caused by the different variety, climatic conditions, cultivation practices, soil, and etc. Although, we would like to drew your attention on a recent review (10.21608/zvjz.2020.22530.1097) which reports that the pumpkin seed possesses a comparatively modest composition of fatty acids, predominantly the essential fatty acids: linoleic, stearic, oleic and palmitic acids, those four fatty acids estimate almost (98 ± 0.13%) of the total amount of fatty acids. However, the text were revised according to you suggestion, please see line 72.

Line 95: … sample collection instead of harvesting.

Authors: This has been done, Please see lines 95.

Table 1: please provide at least for CS and PS the total PUFA and MUFA, the total n-3 and the total n-3, as well as their ratio. This is important and significant information.

Authors: This has been done, Please see Table 1.

Table 2: please add the measurement units.

Authors: This has been done, Please see Table 2.

Line 141: Please explain in words when using first time FAME (fatty acids methyl esters).

Authors: This has been corrected throughout the text.

Table 3: the live weight, is initial or final?

Authors: This has been done, Please see Table 3.

Table 6: Please check the data for C18:3n3, no significant effect? Also for other fatty acids

Authors: All data were re-checked the data for significance. We would like to clarify that all p-values were within the specified thresholds for declaring significance and tendency. As mentioned in lines 170-171, significance was declared at p ≤ 0.05, and tendency was declared at 0.05< p <0.10. We hope this explanation adequately addresses the reviewer's concern. Please let us know if there or clarifications needed.

In the discussion part, some parameters deserve more considerations to be discussed, especially Nitrogen balance. The conclusion is not accurate. According to the data presented in table 6, the C18:3n3 is not significantly increased, only a tendency, this do not count as a major result.

Authors: Thank you for your feedback. We acknowledge that there is a room for more discussion for nitrogen balance. We gave more consideration to this section. Please see lines 233-250.

Additionally, we made note of your concerns in conclusion. Thanks for bringing this to my attention. Please see lines 362-363.

Response to Reviewer 2 Comments:

Reviewer #2: The overall idea of this kind of studies is novel and worth's trying, it helps the livestock producers to find energy sources that may improve the quality of animals products such as milk. There are some issues must be addressed by the authors in order to improve the manuscript.

Authors: Thank you for your valuable feedback on our manuscript. We appreciate your comments and suggestions, and we have carefully considered your feedback in preparing our revised manuscript.

Line 33: delete "DM"

Authors: This has been removed as suggested.

Line 34: please delete "LW"

Authors: This has been removed as suggested.

Line 35: delete "DMI"

Authors: This has been removed as suggested.

Line 36: delete "NDF".

Authors: This has been removed as suggested.

Line 37: delete "ADL" and "FCM".

Authors: This has been removed as suggested.

Line 38: delete "FE"

Authors: This has been removed as suggested.

Line 41: when p value is 0.818, it is not considered tendency, please rewrite "Moreover,..... (p = 0.818).

Authors: This is a Typo, we apologize for this oversight. Please see line 41.

Line 43: according to the results of FA composition in the milk, this statement is not true or at least not enough to draw this conclusion, please rewrite the conclusion statement.

Authors: This has been corrected, Please see line 43.

Line 61-62: how much Mexico is producing from these two seed per year?

Authors: This has been done, Please see lines 62-64.

Line 67-68: Please add their nutrient content in values

Authors: This has been done, Please see lines 67-69.

Table 2: please add the units of the ingredients and chemical composition. In the footnote, please define the diets

Authors: This has been done, Please see Table 2.

Line 162: don't start any statement with abbreviations

Authors: This has been done see line 162.

Table 3: the digestibility of ADF and ADL *is not accurate please re-visit their actual values in check

Authors: We appreciate your attention to detail and the opportunity to address these issues. We have thoroughly reviewed the data, calculations, and spreadsheets. the potential errors should be related to the FAD and measurement techniques. Considering these concerns, we have made the decision to remove the affected data from our analysis. Unfortunately, this means that we no longer have any remaining residue to further analyze.

Table 4: the values of nitrogen balance are questionable, please re-check them. The N balance is too low

Authors: This has been done, the data and calculations were re-checked and the table corrected accordingly, Please see Table 4.

Table 5: please recheck. for example milk composition (g/d) are all wrong

Authors: This has been done, the data and calculations were re-checked and the table corrected accordingly, Please see Table 5.

Although the discussion section is well written, it has to be rewritten to reflect the previous comments regarding some table (see previous comments).

Authors: This has been done, please see lines 178-180, 184-186, and 233-243. 

Response to Reviewer 3 Comments:

Reviewer #3: When I came across the nutrient consumption table, I noticed a fact that will have to be reviewed for me to continue the review:

Let's take the CON treatment as an example:

Table 3 describes:

Forage intake: 1078 g

Concentrate intake: 1386 g

DM intake: 2464g

Corn silage ratio: 1078/2464 = 456.14 g/kg (a value well above the 350 g/kg described in table 2)

NDF intake: 1061 g

DM intake: 2464g

NDF proportion in the diet effectively consumed:

1061/2464 = 456.14 g/kg, a value very, very different from that of table 2 (259 g/kg).

If we do the same thing for EE consumption, we will see that the diet effectively consumed showed values much higher than those described in Table 2.

Crude protein intake was not presented.

Digestibility coefficients draw a lot of attention:

Digestibility of ADF greater than NDF.

ADL digestibility of 430 g/kg, never seen before.

So, to follow the correction, I suggest that the calculations, spreadsheets, etc., be reviewed.

Authors: We sincerely appreciate your valuable input and the opportunity to rectify these issues. We have carefully re-evaluated the data and calculations as per your suggestion. Upon review, we have identified an error in the calculations for ADF and ADL. Considering these concerns in data, we have removed these affected data from our analysis. Furthermore, we acknowledge your observation regarding silage consumption. We have thoroughly verified the data and recalculated the consumption of both silage and concentrate based on the revised information. Thank you once again for your time and attention.

---

## [Decision Letter · Decision Letter 1]

4 Jan 2024

PONE-D-23-22942R1Inclusion of chia seeds (Salvia hispanica L.) and pumpkin seeds (Cucurbita moschata) in dairy sheep dietsPLOS ONE

Dear Dr. Vargas-Bello-Pérez,

Thank you for submitting your manuscript to PLOS ONE. After careful consideration, we feel that it has merit but does not fully meet PLOS ONE’s publication criteria as it currently stands. Therefore, we invite you to submit a revised version of the manuscript that addresses the points raised during the review process.

We look forward to receiving your revised manuscript.

Kind regards,

Aziz ur Rahman Muhammad

Academic Editor

PLOS ONE

Journal Requirements:

Reviewers' comments:

Reviewer's Responses to Questions

**Comments to the Author**

1. If the authors have adequately addressed your comments raised in a previous round of review and you feel that this manuscript is now acceptable for publication, you may indicate that here to bypass the “Comments to the Author” section, enter your conflict of interest statement in the “Confidential to Editor” section, and submit your "Accept" recommendation.

Reviewer #1: All comments have been addressed

Reviewer #2: All comments have been addressed

2. Is the manuscript technically sound, and do the data support the conclusions?

Reviewer #1: Yes

Reviewer #2: Yes

3. Has the statistical analysis been performed appropriately and rigorously? 

Reviewer #1: Yes

Reviewer #2: Yes

4. Have the authors made all data underlying the findings in their manuscript fully available?

Reviewer #1: Yes

Reviewer #2: Yes

5. Is the manuscript presented in an intelligible fashion and written in standard English?

Reviewer #1: Yes

Reviewer #2: No

6. Review Comments to the Author

Reviewer #1: The authors addressed all the questions and observations.

The paper has been reviewed in a nice manner and the corrections are highlighted accordingly.

No further observations.

Reviewer #2: Authors have addressed all comments arisen in the previous revision. However,, a few comments should be addressed in this version of the paper.

Line 66: replace "30-40 g" with "30-40 g/100g"

line 67: replace "30-34 g" with "30-34 g/100g"

Line 59-70: please add some information about one or two references about the use of chia seeds in the livestock diets

Lines 66-67 or 71 or elsewhere: please be consistent , you either use g/100 g or %

Lines 74-75: move it to the previous section of chia seeds

Lines 107, 125 or elsewhere in the manuscript: don't start any sentence with abbreviations

Line 131: please provide full information about the mikoscan analyzer such as the version number, company name, country......

In Table 3: where is the nutrient digestibility for the other nutrients?

In Table 4: still believe that the N lost in the feces and urine is too high and the retained N is too low, please clarify

Line 368: 7%!!! I believe it is 6.1%

7. PLOS authors have the option to publish the peer review history of their article (what does this mean?). If published, this will include your full peer review and any attached files.

Reviewer #1: No

Reviewer #2: No

---

## [Author Response · Author response to Decision Letter 1]

22 Jan 2024

Response to Reviewers

Authors: Thank you for your valuable feedback on our manuscript. We appreciate your comments and suggestions, and we have carefully considered your feedback in preparing our revised manuscript.

Reviewer #2: Authors have addressed all comments arisen in the previous revision. However, a few comments should be addressed in this version of the paper.

Line 66: replace "30-40 g" with "30-40 g/100g"

Authors: This has been done, Please see line 79

line 67: replace "30-34 g" with "30-34 g/100g"

Authors: This has been done, Please see line 80

Line 59-70: please add some information about one or two references about the use of chia seeds in the livestock diets

Authors: This has been done, Please see lines 66 - 78

Lines 66-67 or 71 or elsewhere: please be consistent, you either use g/100 g or %

Authors: This has been done, Please see line s79-80, 84

L84 seed has a crude protein content of approximately 35 g/100g and from 30 to 50 g/100g of oil [2]. Changed accordingly 

Lines 74-75: move it to the previous section of chia seeds

Authors: This has been done, Please see lines 66 -78

Lines 107, 125 or elsewhere in the manuscript: don't start any sentence with abbreviations

Authors: This has been done, Please see linen 119

Line 131: please provide full information about the mikoscan analyzer such as the version number, company name, country......

R Authors: This has been done, Please see line 143

 SL60. Milkotronic LTD. Nova Zagora, Bulgaria

In Table 3: where is the nutrient digestibility for the other nutrients?

Authors: In the past revision we explained the following regarding missing nutrient digestibility data: We sincerely appreciate your valuable input and the opportunity to rectify these issues. We have carefully re-evaluated the data and calculations as suggested. Upon review, we have identified an error in the calculations for ADF and ADL. Considering these concerns regarding the data, we have removed all affected data from our analysis.

In Table 4: still believe that the N lost in the feces and urine is too high and the retained N is too low, please clarify

Authors: We have reviewed the data and they are correct, especially because they are producing milk and in this case the greatest loss of N was in urine, we reviewed the data and they are correct, and when we make the correction of N in milk is when we get negative. That would be normal for a cow or ewe at the beginning of lactation in many cases.

Line 368: 7%!!! I believe it is 6.1%

Authors: This has been done, Please see line 380

You are right, this data was corrected in the previous revision, and we did not adjust it in the conclusions, thank you for your observation.

---

## [Editor Report · Decision Letter 2]

1 Feb 2024

PONE-D-23-22942R2Inclusion of chia seeds (Salvia hispanica L.) and pumpkin seeds (Cucurbita moschata) in dairy sheep dietsPLOS ONE

Dear Dr. Vargas-Bello-Pérez,

Thank you for submitting your manuscript to PLOS ONE. After careful consideration, we feel that it has merit but does not fully meet PLOS ONE’s publication criteria as it currently stands. Therefore, we invite you to submit a revised version of the manuscript that addresses the points raised during the review process.

Although you have addressed major issues of the current manuscript, however, there are few changes especially grammar and some numerical values need to be addressed. Authors need these to be addressed before publication. Authors are advised to respond and address the said comments

We look forward to receiving your revised manuscript.

Kind regards,

Aziz ur Rahman Muhammad

Academic Editor

PLOS ONE

Additional Editor Comments :

Dear Authors

Although you have addressed major issues of the current manuscript, however, there are few changes especially grammar and some numerical values need to be addressed. Authors need these to be addressed before publication. Authors are advised to respond and address the said comments

---

## [Author Response · Author response to Decision Letter 2]

12 Feb 2024

Dear Authors

Although you have addressed major issues of the current manuscript, however, there are few changes especially grammar and some numerical values need to be addressed. Authors need these to be addressed before publication. Authors are advised to respond and address the said comments

AUTHORS

We have made English style corrections and double-checked numbers. All changes have been done in red colour.

---

## [Editor Report · Decision Letter 3]

19 Feb 2024

PONE-D-23-22942R3Inclusion of chia seeds (Salvia hispanica L.) and pumpkin seeds (Cucurbita moschata) in dairy sheep dietsPLOS ONE

Dear Dr. Vargas-Bello-Pérez,

Thank you for submitting your manuscript to PLOS ONE. After careful consideration, we feel that it has merit but does not fully meet PLOS ONE’s publication criteria as it currently stands. Therefore, we invite you to submit a revised version of the manuscript that addresses the points raised during the review process.

Dear Authors

I am unable to find detailed author response on the reviewer comments. Furthermore, I am also unable to find color changes in the manuscript as authors claimed. Therefore, I would like to send it back to author for revision. 

We look forward to receiving your revised manuscript.

Kind regards,

Aziz ur Rahman Muhammad

Academic Editor

PLOS ONE

Dear Editor

I am unable to find author response on the reviewer comments. Furthermore, i am also unable to find color changes in the manuscript as authors claimed. Therefore, i would like to send it back to author for revision.

---

## [Author Response · Author response to Decision Letter 3]

19 Feb 2024

REBUTTAL LETTER

Dear Authors

Although you have addressed major issues of the current manuscript, however, there are few changes especially grammar and some numerical values need to be addressed. Authors need these to be addressed before publication. Authors are advised to respond and address the said comments

AUTHORS

We have made English style corrections and double-checked numbers. All changes have been done in red colour.

---

## [Decision Letter · Decision Letter 4]

7 Mar 2024

Inclusion of chia seeds (Salvia hispanica L.) and pumpkin seeds (Cucurbita moschata) in dairy sheep diets

PONE-D-23-22942R4

Dear Dr. Vargas-Bello-Pérez,

We’re pleased to inform you that your manuscript has been judged scientifically suitable for publication and will be formally accepted for publication once it meets all outstanding technical requirements.

Kind regards,

Aziz ur Rahman Muhammad

Academic Editor

PLOS ONE

Additional Editor Comments (optional):

Dear Authors

Thanks for addressing the comments of reviewers. congrats

Reviewers' comments:

Reviewer's Responses to Questions

**Comments to the Author**

1. If the authors have adequately addressed your comments raised in a previous round of review and you feel that this manuscript is now acceptable for publication, you may indicate that here to bypass the “Comments to the Author” section, enter your conflict of interest statement in the “Confidential to Editor” section, and submit your "Accept" recommendation.

Reviewer #2: All comments have been addressed

2. Is the manuscript technically sound, and do the data support the conclusions?

Reviewer #2: Yes

3. Has the statistical analysis been performed appropriately and rigorously? 

Reviewer #2: Yes

4. Have the authors made all data underlying the findings in their manuscript fully available?

Reviewer #2: Yes

5. Is the manuscript presented in an intelligible fashion and written in standard English?

Reviewer #2: Yes

6. Review Comments to the Author

Reviewer #2: No more comments, the authors addressed all comments arisen in the previous version. I recommend accepting the manuscript in the current form

7. PLOS authors have the option to publish the peer review history of their article (what does this mean?). If published, this will include your full peer review and any attached files.

Reviewer #2: No

---

## [Editor Report · Acceptance letter]

27 Mar 2024

PONE-D-23-22942R4 

PLOS ONE

Dear Dr. Vargas-Bello-Pérez, 

I'm pleased to inform you that your manuscript has been deemed suitable for publication in PLOS ONE. Congratulations! Your manuscript is now being handed over to our production team.

Kind regards, 

on behalf of

Dr. Aziz ur Rahman Muhammad 

Academic Editor

PLOS ONE